# Synthesis and Electrochemical Properties of Bi_2_MoO_6_/Carbon Anode for Lithium-Ion Battery Application

**DOI:** 10.3390/ma13051132

**Published:** 2020-03-04

**Authors:** Tingting Zhang, Emilia Olsson, Mohammadmehdi Choolaei, Vlad Stolojan, Chuanqi Feng, Huimin Wu, Shiquan Wang, Qiong Cai

**Affiliations:** 1Hubei Collaborative Innovation Center for Advanced Organic Chemical Materials and Ministry of Education Key Laboratory for Synthesis and Applications of Organic Functional Molecules, Hubei University, Wuhan 430062, China; zhangtingting@iccas.ac.cn (T.Z.); whm267@126.com (H.W.); wsqhao@hubu.edu.cn (S.W.); 2Depatment of Chemical and Process Engineering, Faculty of Engineering and Physical Sciences, University of Surrey, Guildford GU2 7XH, UK; k.olsson@surrey.ac.uk (E.O.); m.choolaei@surrey.ac.uk (M.C.); 3Advanced Technology Institute, Department of Electrical and Electronic Engineering, Faculty of Engineering and Physical Sciences, University of Surrey, Guildford GU2 7XH, UK; V.Stolojan@surrey.ac.uk

**Keywords:** Li ion batteries, composite, electrode materials, hydrothermal synthesis, electrochemical performance, ab initio calculations

## Abstract

High capacity electrode materials are the key for high energy density Li-ion batteries (LIB) to meet the requirement of the increased driving range of electric vehicles. Here we report the synthesis of a novel anode material, Bi_2_MoO_6_/palm-carbon composite, via a simple hydrothermal method. The composite shows higher reversible capacity and better cycling performance, compared to pure Bi_2_MoO_6_. In 0–3 V, a potential window of 100 mA/g current density, the LIB cells based on Bi_2_MoO_6_/palm-carbon composite show retention reversible capacity of 664 mAh·g^−1^ after 200 cycles. Electrochemical testing and *ab initio* density functional theory calculations are used to study the fundamental mechanism of Li ion incorporation into the materials. These studies confirm that Li ions incorporate into Bi_2_MoO_6_ via insertion to the interstitial sites in the MoO_6_-layer, and the presence of palm-carbon improves the electronic conductivity, and thus enhanced the performance of the composite materials.

## 1. Introduction

Lithium-ion batteries (LIBs) are ubiquitous in electric vehicles, laptops, mobile phones and various electronic products for energy storage, due to their high energy density, good electronic performance, low self-discharge and long cycle life [1,2,3]. The properties of the electrode and electrolyte materials determine the electrochemical performance of a LIB cell, and thus have been researched widely. Negative electrode (i.e., anode) materials with low operating potentials (close to 0 V_Li_ (Li^+^/Li)) and high storage capacity are important for achieving high LIB battery performance, as the overall cell voltage is determined by the difference between the positive electrode and the negative electrode. Commercial LIBs normally utilize a graphite anode which operates at 0.1 V_Li_ with good stability in conventional liquid carbonate electrolytes and give a specific capacity of 372 mAh·g^−1^ [4,5,6,7,8]. To further increase the driving range of electric vehicles and the lifetime of portable electronics between charge and discharge, we need to have much higher capacity—at least double the capacity of graphite anodes used in LIBs. Thus, there has been development of new anode materials. Lithium metal has been considered as a promising anode material for LIBs owing to its ultra-high theoretical capacity (3860 mAh g^−1^) [9], but face significant challenges in the uncontrolled dendrite formation and the associated huge performance degradation. Although the promise of Li metal anode has attracted a lot of research interests, with considerable advances being made into understanding the failure mechanisms and various strategies to mitigate the Li dendrite formation, it is still far from practical utilizations in commercial LIBs [9,10,11]. The alloy anode materials such as tin and silicon show high specific capacity (1000–4000 mAh·g^−1^) and low working voltage (0.1–1.0 V_Li_). However, the expansion of volume (as much as 400%) and the consequent irreversible morphological and mechanical changes lead to a significant decrease of capacity during the alloy reaction, which prohibit the development of commercial products. Although recent advances in nanostructure, carbon coating and Si/Sn based composite have improved the stability and the electrochemical performance of the anode materials, further development is still needed [5,7,8,12]. Conversion type transition metal oxides (such as oxides of iron, manganese, cobalt, copper and nickel) [13] show high specific capacities and high rate capabilities, and have attracted increased attention. However, low electrical conductivity [14] and the fragility of the electrode limit their development and wider use in practical applications [3,5,6,7,8]. 

Both binary oxides (MOx, M = Ti, Mn, Sn, Fe, Co, Mo, Ni and Cu, etc.) [14,15,16] and ternary oxides (ABOx A = Ca, Mg, Bi, Ti, Zn, Fe, V, Mn, Co, Ni, Cu; B = Mn, Mo, Co, Fe, Ni, Ti, Nb, Cu, Sn; A≠B.) [17,18,19,20,21,22,23,24] based on different reaction mechanisms (e.g., intercalation, conversion, and alloy reactions) have been proposed as electrode materials. Ternary oxides are more versatile, with a great number of possible combinations of metal A and B. ABOx with nanostructure are advantageous as they have a high specific surface area, as well as a shorter diffusion path for Li ion than their micro-sized or bulk counterparts. However, most of the nano-sized materials exhibit rapid capacity degradation due to the inherent nature of low conductivity and aggregation over cycling [13,14,15,16]. To increase the electrochemical performance of metal oxides as battery electrodes, various approaches have been investigated, including: (1) minimizing particle size and optimizing particle shape, thickness, and nanostructure self-assembly [8,17], (2) fabricating hierarchically porous structures [25] in order to enhance the buffer space and active sites of electrochemical reaction which improve the rate capability and cycling stability of batteries, (3) combining hybrid metal oxides with large surface area and fast electron transport materials [26], in which case, various carbon materials have been studied for their high surface area, proper morphology and structures, and excellent electron transfer properties [12,27]. Porous carbonaceous materials [12,27] with tunable porosities, including activated carbon, ordered mesoporous carbons, carbon aerogels and graphene-based materials, have been widely used in the field of electrode materials in batteries. Carbon materials derived from biomass have drawn much interest, as they are sustainable, low cost, and give excellent properties such as high specific surface area, well-developed porous structure, high electrical conductivity, and electrochemical stability [27,28]. 

Many molybdenum based oxides were widely used as anode materials because of their high specific capacity, multiple valance states of oxides and high mass density which increased the capacity of the electrode materials. Many Mo based oxides, such as binary oxides (MoO_3_, MoO_2_), ternary oxides (CaMoO_4_, MMoO_4_, M = Ni, Co, Mn) and CoMoO_4_ were investigated as anode materials for LIBs [20,21,22,23,24]. However, unfortunately, complex preparation methods hinder their commercial application. Metal molybdates, particularly bismuth molybdate (Bi_2_MoO_6_), which is a conventional layered metal molybate consisting of [Bi_2_O_2_]^+^ layers sandwiched between [MO_4_]^2-^ slabs, could be used as potential LIB anode materials. The limited research on Bi_2_MoO_6_ as LIB anode shows that Bi_2_MoO_6_ has a much higher storage capacity than graphite, with initial charge capacity of over 900 mAh·g^−1^ reported [25,26], and can be considered as a promising material for an LIB anode. However, in the reported work [25,26,29,30,31], complicated processes and expensive materials (such as graphene and Ni foam) have been taken to synthesize Bi_2_MoO_6_ based materials, which are expensive and not suitable for large scale practical applications. Based on previous experience with metal oxides [14,15,16,17,18,19], the electrochemical performance of Bi_2_MoO_6_ could be improved by hybridizing Bi_2_MoO_6_ with cheap carbon materials, using simpler and cost-effective synthesis methods. 

In this paper, we report our work on the development of Bi_2_MoO_6_ hybridized with carbon as LIB anode materials, via a simple hydrothermal route. The carbon material in this paper is synthesized from palm tree leaves, and is hereafter referred to as palm carbon. Palm carbon is chosen because it is easy to obtain, it shows a high electronic conductivity and is derived from a more sustainable source than other carbon materials, such as graphene and those derived from polymers. The prepared materials are characterized using a range of techniques including XRD, SEM, STEM (Scanning Transmission Electron Microscopy), Raman and XPS to understand the morphology and chemical properties. Density functional theory (DFT) calculations are also performed to gain insights into the structure of Bi_2_MoO_6_ and Li insertion process in Bi_2_MoO_6._ The as-synthesized materials are then tested in LIB cells. The results show that the hybrid Bi_2_MoO_6_/palm carbon materials give a much improved performance compared to Bi_2_MoO_6_. 

## 2. Experimental and Computational Details

### 2.1. Materials Synthesis 

All chemical reagents in our study were analytical grade. The Bi_2_MoO_6_ material was synthesized by hydrothermal methods. In addition, 2 mmol Bi(NO_3_)_3_ and 1 mmol Na_2_MoO_4_ were dissolved in 20 mL de-ionized water under magnetically stirring for 1 h (h) to form a uniform mixture solution of bismuth molybdate. The mixture solution was then put in a 50 mL hydrothermal reactor and heated by an electric oven at 180 °C for 12 h. The produced precipitates were centrifuged and washed with water and ethanol for several times, and dried under vacuum at 60 °C. Finally, the pure Bi_2_MoO_6_ was obtained, denoted as BMO.

The palm raw material was first washed with de-ionized water and dried under air at 60 °C. The precursor was then boiled in saturated sodium hydroxide (NaOH) for 6 h at 110 °C and soaked in H_2_O_2_ (30%) for 12 h at 60 °C. Finally, the obtained precursor was centrifuged and washed with de-ionized water and ethanol, and dried under vacuum at 60 °C. The as-prepared palm was pre-oxidized in a tube furnace at 350 °C for 2 h in oxygen atmosphere. The pre-oxidized sample was then carbonized in a tube furnace at 550 °C for 3 h in argon atmosphere, at a heating rate of 3 °C∙min^−1^. 

The as-prepared palm carbon was ultrasonically dispersed in 10 mL de-ionized water for 2 h; the palm carbon solution was then added dropwise to the Bi/Mo mixture, under constant stirring. After one hour of stirring, the mixed solution was transferred into a 50 mL Teflon-lined stainless-steel autoclave and heated in an oven at 180 °C for a duration of 12 h. The obtained precursor was centrifuged and washed with water and ethanol several times, and dried under vacuum at 60 °C. The powders obtained were annealed in a tube furnace with a temperature ramp of 1 °C∙min-1 at 550 °C for 2 h, to yield the sample denoted as BMO/C. The weight percentage of palm carbon in the as-synthesized BMO/C sample is 2%, giving a weight ratio of metal oxide to carbon as 49:1.

### 2.2. Materials Characterization 

The crystal structure of the BMO/C was characterized by X-ray diffraction (XRD) using Cu Kα radiation (λ = 0.15406 nm) under a voltage of 40 KV and a current of 40 mA. Scanning electron microscopy (SEM; JEOL JSM, 6510 V) and scanning transmission electron microscopy (STEM, Hitachi HD2300A, Tokyo, Japan, operated at 200 keV—Schottky field emission gun) are used to characterize the morphology of the prepared compounds. Samples were prepared by dispersing the powder in isopropanol and sonicating them for about 20 min. A drop of solution was filtered through a holey-carbon grid; the grid was dried at 100 °C for 5 min before being transferred to the microscope. Raman spectroscopy of the synthesized powders were conducted using Renishaw 2000 (514 nm green laser, 40 mW) over the range of 100–1000 cm^−1^. The oxidation states of the samples were investigated using an X-ray photoelectron spectrometer (XPS, Escalab 250Xi, Massachusetts, MA, USA). 

### 2.3. Electrochemical Measurements

The charge and discharge tests were tested by a CR2025 button cell on the battery testing system (Neware, Shengzhen, China). Two working electrodes were prepared, using BMO and BMO/C respectively, as active materials. The working electrodes were made of active material, acetylene black, and Carboxymethylcellulose sodium (CMC) with a molar ratio of 7:2:1. The electrodes were dried at 100 °C in a vacuum furnace for overnight. The separator was Celgard 2400 porous polypropylene. The electrolyte, LiPF_6_ (1 mol·L^−1^), was mixed with ethylene carbonate (EC) and diethyl carbonate (DEC) at a volume ratio of 1:1. Li metal was used as the counter electrode. All the tests were assembled in an argon-filled box containing less than 1 ppm each of oxygen and moisture. The charge and discharge processes were tested at a constant current density of 100 mA/g and a voltage range of 0.01 to 3.00 V. The typical mass of the electrode material used in the experimental ranged from 5 to 8 mg. Electrochemical impedance spectroscopy (EIS) experiments and cyclic voltammetry (CV) were conducted using a CHI 600 E electrochemical workstation.

### 2.4. Computational Details

Density functional theory (DFT) calculations were conducted in the Vienna Ab initio Simulation Package (VASP, version 5.3.5) [32,33,34,35] to elucidate the Li insertion into Bi_2_MoO_6._ To describe the ion–electron interaction, the projector-augmented wave method (PAW) was used [36]. Based on convergence tests, the plane wave cut-off and k-space integrals were chosen so that the total energy was converged to 1 meV/atom, the kinetic energy cut-offs for all systems were set to 600 eV, with a 6 × 3 × 6 Γ-centered Monkhorst–Pack grid to sample the Brillouin zone [37]. The tetrahedron method with Blöchl corrections for smearing [36,37,38] was further applied. The generalized gradient approximation (GGA) with Perdew–Burke–Ernzerhof (PBE) [39,40] functionals were used to describe the interacting electron exchange–correlation energy, with an electronic convergence criteria of 10^−5^ eV and an ionic convergence criteria of 10^−3^ eV·Å^−1^. All of the calculations were performed spin-polarized. Bader AIM (Atoms in Molecules) charges [41] were calculated with the Henkelman algorithm [42]. Bi_2_MoO_6_ is experimentally seen to be a semi-conductor. It is well known that uncorrected GGA underestimates the band gaps of strongly correlated systems due to the DFT electron self-interaction error, and hence we have used the On-site Coulombic interaction (DFT + U) for the Mo *d*-electrons to account for this error [43,44,45,46], by means of Dudarev’s approach [47]. Hubbard parameters (*U_eff_*) of 8.6 eV for Mo previously parametrized by Getsoian et al. were used [46]. Due to the large polarizability of this Mo upon Li insertion [47,48,49,50], the DFT-D3 method with Becke–Jonson damping of Grimme and co-workers [51] was included, as has been successfully applied for this system elsewhere [31]. 

## 3. Results and Discussion

### 3.1. Materials Characterization

#### Computational Characterization of Bi_2_MoO_6_

To evaluate energy storage technologies, experimental observations combined with atomistic insights provided by theory (DFT) is an extremely powerful tool. Knowledge of the electronic states, relative occupations, magnetic moments, and oxidation/reduction behavior allows an understanding of the electrochemistry of the system to be elucidated. This is then applied to both the pristine and the defective (here Li interstitials or Li substitutional defects) Bi_2_MoO_6_ systems, giving atomistic meaning to the experimental Li storage results. Here, firstly, the pristine Bi_2_MoO_6_ structure before Li addition is studied to understand the atomic scale structure of our anode material, and the effect of Li storage in Bi_2_MoO_6_ is included in Section 3.4.

Three crystal structures of Bi_2_MoO_6_ have been investigated: orthorhombic with space group Pca2_1_ (Figure 1a), monoclinic P21/c (Figure 1b), and Pbca (Figure 1c). Based on the total energy from cell optimizations of these cells, the orthorhombic Pca2_1_ is found to be the most stable structure of Bi_2_MoO_6_, which is in agreement with experiment [26]. It is worth noting that, at higher temperatures, a phase transition from the orthorhombic to monoclinic phase has been observed experimentally [52]. For the application of Bi_2_MoO_6_ as an anode material for LIBs, the low temperature orthorhombic structure (as shown in Figure 1a) will be used. 

Orthorhombic Bi_2_MoO_6_ has a layered structure, with alternating layers of corner-sharing MoO_6_ octahedra and Bi-O-Bi layers. As discussed in the computational details, non-corrected GGA calculations can result in an underestimation of the electronic band gap. The geometric and electronic structures of orthorhombic Bi_2_MoO_6_ were optimized with both GGA and GGA + U to understand these differences, as shown in Table 1. It was found that the band gap calculated with GGA underestimated the previously reported experimental band gap by 0.3 eV, whereas including a U-correction gave a band gap of 2.45 eV, which is closer to the measured values of 2.53 eV [53], and 2.56 eV [54], respectively. The GGA + U method also achieved a better agreement with experimental data in terms of lattice parameters, as GGA was found to overestimate the short (a and c) lattice vectors. The obtained lattice parameters with GGA were a = 5.66 Å, b = 16.49 Å, and c = 5.68 Å, with GGA + U giving *a* = 5.45 Å, *b* = 16.51 Å, and *c* = 5.47 Å, as compared to experimental *a* = 5.45 Å, *b* = 16.47 Å, and *c* = 5.47 Å.

Calculations of the electronic structure of Bi_2_MoO_6_ in terms of the projected density of states (PDOS) (Figure 2) reveal that the valence band maximum consists of O p-states, whereas Mo d-states and the Bi p-states make up the conduction band. Utilizing the Bader charge oxidation state convention presented by Getosian et al. [46] for Bi_2_MoO_6_, Bader charge analysis confirms the experimental findings of assigning an oxidation number of +3 and +6 for bismuth and molybdenum, respectively. Our calculated Bader charges (per the outlined methodology in Section 2.4) are 1.91 for each bismuth ion, 2.76 for each molybdenum ion, and −1.10 for the oxygen ions, which corresponds to a formal oxygen oxidation state of −2.

### 3.2. Experimental Characterization

The crystallographic structures of both BMO and BMO/C samples were examined by X-ray diffraction (XRD). Figure 3a shows all major diffraction peaks appearing at the same lattice planes, at 2*θ*: 10.927°, 28.309°, 32.533°, 32.642°, 36.055°, 46.737°, 47.175°, 55.585°, 56.251°, and 58.477°, corresponding to the lattice planes (2 0 0), (0 2 0), (1 3 1), (0 0 2), (1 5 1), (2 0 2), (0 6 2), (1 3 3), (1 9 1) and (2 6 2) of the Bi_2_MoO_6_ structure with high crystallinity (JCPDS card No. 21-0102). These diffraction peaks were furthermore observed in the simulated BMO XRD pattern from DFT calculations (Figure 3b). No other impurity peaks are observed. By comparison, only strong peaks of BMO/C phase were presented in the diffraction peak indexed to the lattice planes (2 0 0) and (0 2 0) corresponding to the lattices planes at 2*θ*: 10.927° and 32.533°. The difference between the two samples was due to the promoted growth of BMO by palm carbon. In addition, intensity of peaks for BMO/C was so strong that diffraction peaks of carbon was not detected, which is in accordance with the SEM and TEM results. To further investigate the phase of the as-obtained Bi_2_MoO_6_/C, Raman spectroscopy was employed (Figure 3b,c). All the characteristic vibrational bonds for Bi_2_MoO_6_ can be observed in Figure 3b: 845 cm^−1^ (s), 796 cm^−1^ (vs.), 713 cm^−1^ (m), 402 cm^−1^ (m), 352 cm^−1^ (s), 325 cm^−1^ (w), 292 cm^−1^ (m, sh), 282 cm^−1^ (s), 230 cm^−1^ (w), 195 cm^−1^ (m), 139 cm^−1^ (m) [17]. The band at 139 cm^−1^ originates from the lattice modes of Bi^3+^ atoms mainly in the direction perpendicular to the layers [17,26,48]. The 190–405 cm^−1^ range mostly corresponds to the stretching and bending modes of BiO_3_ tetrahedra, coupled with bending motions of the MoO_6_ octahedra [55]; the modes at 282 and 292 cm^−1^ most likely correspond to E_g_ bending vibration and the ones around 325, 352, and 402 cm^−1^ originate from E_u_ symmetry bending modes [56]. The less intense peaks in the range of 400–600 cm^−1^ can be assigned to the stretching modes of the Mo-O bonds and the twisting mode in Bi_2_MoO_6_ [57]_._ As for the modes observed above 700 cm^−1^, the peak at 796 cm^−1^ is attributed to the symmetric stretch of a MoO_6_ octahedron, whereas modes at 714 and 844 cm^−1^ represent the orthorhombic distortions of the MoO_6_ octahedron in Bi_2_MoO_6_ [56]_._ In comparison to BMO, the peaks of BMO/C were found to broaden and slightly shift toward lower wavenumbers [58], in addition to the presence of a small peak at 885 cm^−1^, originating from the vibration of Mo-O bonds in MoO_4_. Nevertheless, the absence of significant spectral deviation from the samples revealed that the main structure of the Bi_2_MoO_6_ phase was not dramatically disturbed. To further understand the nature of carbon materials in the BMO/C sample, the sample was examined in the 1000–2000 cm^−1^ wave number range. Figure 3c shows the Raman spectra of the sample with two peaks appearing at 1353 cm^−1^ (D-band, disordered carbon) and 1589 cm^−1^ (G-band, graphitic carbon). The D-band represents the disordered carbon with defect (i.e., the broken of the 6-member ring symmetry) and the G-band corresponds to the sp^2^ hybridized carbon with graphitic carbon (i.e., with the perfect 6-member ring symmetry). The intensity ratio of D-band and G-band (I_D_/I_G_) gives an indicator of the degree of disorder of the carbon. Figure 3c shows the intensity ratio of I_D_/I_G_ = 0.85, indicating that the sample of BMO/C is present with both graphitic carbon and disordered carbon. 

The morphology of the as-synthesized BMO was characterized using SEM and STEM. Figure 4a,c show the SEM images of BMO particles, with a particle length of about 1 μm. Figure 4b,d show the SEM images of BMO/C particles of 3D nanosheets, with a particle length of 100 nm. Upon modification with palm carbon, the flakes appear to be shortened and aggregated, becoming more rounded in nature, but of smaller sizes. Figure 4e,f show the STEM images of BMO and BMO/C, respectively. The BMO/C composite image is obtained by overlaying the High-Angle Annular Dark Field image, whose contrast is proportional to the Z^2^ of the material, on top of the Secondary Electron image, which reveals the surface morphology of the sample. It also shows that the Bi_2_MoO_6_ particles are attached to the palm carbon scrolls (green color), confirming the composite nature of the material. This is because the model approximates hydrogenic cross-sections and flat samples, whilst the experimental comparison is between the Bi M-edge and the Mo L-edge; XRD confirms that the crystal structure remains that of the stoichiometric BMO. Figure 4g,h show the statistical analysis of long/short axes ratio for the Bi_2_MoO_6_ particles in the BMO and BMO/C samples, with an average value around 4.45 for the BMO sample and 1.19 for the BMO/C sample. This indicates that the BMO sample shows three-dimensional slender needles; however, upon modification with palm carbon, the Bi_2_MoO_6_ particles appear to shorten, becoming more rounded in nature. Figure 4i,j show the statistical analysis of the Bi_2_MoO_6_ particle size in the BMO and BMO/C samples. It is clear that the particles are much smaller in the BMO/C sample (with an average particle size of ~482 nm) than those in the BMO sample (with an average particle size of ~154 nm). We conclude that the supply of the palm carbon has a remarkable effect on the morphology of the BMO quasi-nanometer microspheres. 

The chemical compositions and the surface electronic states of the BMO/C sample are investigated using XPS, with results shown in Figure 5. All the expected elements including Bi, Mo, C and O can be found in the survey spectra. As shown in Figure 5b, two binding energy peaks appeared at 164.6 eV for Bi 4f_7/2_ and 159.3 eV for Bi 4f _5/2_, revealing that Bi is in the Bi^3+^ oxidation state. Figure 5c shows two peaks with binding energies of 235.8 and 234.2 eV which are assigned to the Mo^6+^ ions, respectively. The peak of the O 1s spectrum was observed at 530.27 eV, which is attributed to the lattice oxygen and near surface oxygen in bismuth molybdate [25,26]. The analysis of XPS shows two obvious peaks at 284.5 eV and 286.2 eV, corresponding to C-C sp^2^, and C-C sp^3^, respectively. A slight O=C=O peak at 288.6 eV is observed, due to the pre-oxidization of the sample [26]. The XPS results confirm the composition of the BMO/C sample as inferred from EDX and XRD and show that the surface is oxygen deficient, which is expected to help improve the performance.

### 3.3. Electrochemical Properties

Figure 6a,b show the discharge and charge curves of BMO and BMO/C electrodes during 1st, 2nd and 20th cycles at a constant current density of 100 mA·g^−1^ and a voltage range of 0.01 to 3.0 V (vs. Li^+^/Li). It can be seen from Figure 6a that BMO features two plateaus at approximately 0.65 V and 0.75 V; the initial discharge and charge specific capacity of the BMO electrode are 841 and 753 mAh∙g^−1^, with an initial columbic efficiency of 89% (which is the ratio of the first cycle discharge capacity to the first cycle charge). From Figure 6a, we can infer that the reaction during the first discharge involves intercalation of Li into the BMO lattice, followed by the destruction of the crystal structure, the amorphization of the BMO lattice, and finally the formation of Bi and Mo metals [20]. Figure 6b shows that BMO/C also features two plateaus at about 0.5 and 0.6 V which are less than the plateaus of BMO, and the initial discharge and charge specific capacity of the BMO/C electrode are 1014 and 889 mAh∙g^−1^, with an initial columbic efficiency of 87%. The enhanced performance of BMO/C can be attributed to two factors: (1) the more rounded shape of the BMO particles, providing more reacting surface sites [59,60,61,62]; (2) palm carbon materials that improve the ionic and electronic transport and make more BMO reaction sites accessible hence increase the storage capacity. 

Figure 6c shows the cycling performance and the columbic efficiency of the BMO and the BMO/C electrodes. The cycling performance of the BMO/C electrode is superior to the BMO electrode. The specific capacities of both electrodes decrease after 20 cycles, with the BMO/C electrode maintaining a capacity of 731 mAh·g^−1^ and the BMO electrode retaining a capacity of 580 mAh∙g^−1^. At a current density of 100 mA/g, the loss of the second cycle discharge specific capacity of BMO/C is 8% less than that of BMO. Remarkably, the increase of the capacity is attributable to the fragmentation of BMO/C particles during the cycling process, which increases the contact of the material with the electrolyte and thus the specific capacity. This is similar to the behavior of transition metal oxide anodes such as Co_3_O_4_ [60,61] and ZnMn_2_O_4_ [62]. The BMO/C electrode achieved a high reversible capacity of 664 mAh∙g^−1^ after 200 cycles, while the BMO electrode only obtained a capacity of 282 mAh∙g^−1^ after 200 cycles. The BMO/C electrode in this work shows improved cycle performance compared to the reported Bi_2_MoO_6_/reduced graphene oxide composites, which has a retention capacity of 705 mAh·g^−1^ after 100 cycles [26]. Figure 6d shows the capacity retention at load current densities of 100, 200, 500 and 1000 mA∙g^−1^; the discharge specific capacities of the BMO/C electrode are 501, 426, 380 and 332 mAh∙g^−1^. The specific capacity could be recovered to 547 mAh∙g^−1^ when the load current density is returned to 100 mA∙g^−1^. The BMO electrode, however, gives a much lower specific capacity under the same condition. The specific capacities of the BMO electrode measured at the same current densities are 436, 361, 320 and 283 mAh∙g^−1^. With the current density being brought back to 100 mA∙g^−1^, the retained capacity is 484 mAh∙g^−1^. The better electrochemical properties of BMO/C could be ascribed to the presence of palm carbon [63,64], which facilitates the charge transfer process in the BMO/C layers. In addition, the fragmentation of the BMO structure during cycling [17,18,19] provides a higher number of smaller BMO particles, and thus higher surface area and access for electrochemical reactions with Li. The reactions of Li with ternary oxides such as Bi_2_MoO_6_ involve multistep reactions which include: intercalation, conversion reactions (i.e., Bi and Mo are electrochemically active with respect to Li) and alloying under further reduction of Bi^3+^ to Bi. Extra discharge capacity can be obtained from the formation of Li–Bi alloy [17,18,19]. These factors lead to a significant improvement of the electrochemical performance for Li-ion batteries [16,18,26]. 

Figure 7a shows the electrochemical impedance spectra (EIS) of the samples acquired in the frequency range of 0.01 and 100 KHz. The EIS measurement was carried out with fresh cells at an open circuit voltage (OCV) of 2.21 eV and 2.60 eV, respectively, for the BMO and BMO/C electrodes. The intercept of the semicircle appearance of the high frequency is attributed to the ohmic resistance (the electrolyte resistance, R_s_) [65] in the equivalent circuit, which is related to the solid electrolyte interphase (SEI) film and the contact resistance. The semicircle in the medium frequency region corresponds to the charge transfer between the electrode and electrolyte. The semicircle is interpreted as a parallel circuit which consists of the double layer capacitance [66], C_dl_, and the charge transfer resistance [67], R_ct_. The inclined lines in the low frequency range are related to the lithium ion diffusion in the BMO and BMO/C electrodes, and are associated with Warburg impedance [68,69] Z_w_ in the equivalent circuit. The resistance of BMO/C electrode of 138 Ω is lower than that of the BMO electrode (with a resistance of 246 Ω). The result indicates that the BMO/C electrode exhibits better electronic conductivity than the BMO electrode. Das et al. reported the electrochemical impedance spectroscopy (EIS) and the Nyquist (Z′ Vs. - Z″) plots of a Li/Co_2_Mo_3_O_8_ system at a current density of 60 mA/g and a voltage range of 0.05–3 V during the first discharge and charge cycle [23]. When the OCV is 2.7 V, the Nyquist plot showed a semicircle in the high frequency range and an arc type in the low frequency range. It was indicated that, when the OCV decreases to 1.5 V, the Nyquist plot had no significant change except a decrease in the overall impedance. When the discharged voltage was close to 0.005 V, the second semicircle appeared in the intermediate frequency region at 0.9 V, which was attributed to the impedance from the bulk. This was consistent with the galvanostatic cycling data, which indicated the destruction of the crystal structure [23]. The impedance values during the first discharge/charge cycle supported the reaction mechanism which had the crystal structure destruction and reaction, as reflected by an increase in the overall impedance value until 0.005 V. When charged, the overall impedance value decreased with the reformation of corresponding metal oxides [20,21,22,23].

Figure 7b shows the linear fitting Warburg impedance of BMO and BMO/C. The Warburg impedance represents the impedance that is generated by the diffusion of Li ions [70] in the lattice of electrode material. According to the formula Z′ = R_s_ + R_ct_ + A_w_ω^−1/2^, A_w_ is related to the slope of the Warburg impedance diagram, and ω is assigned to the angular frequency of the alternating current. The Li^+^ diffusion coefficient in the electrode can be estimated using Equation (1): (1)$$D_{{\mathrm{Li}}^{+}}=0.5[\frac{V_{m}}{{\mathrm{FSA}}_{w}}\left(-\frac{\mathrm{dE}}{\mathrm{dx}}\right){]}^{2}$$
where V_m_ is the molar volume of the material, F is the Faraday constant, S is the apparent surface area of the electrode, and (dE)/(dx) is the slope of the open circuit potentialvs. the mobile ion concentration x at each x value. Based on the molar volume, the electrode surface area and (dE)/(dx) were substantially the same. The Li ion diffusion coefficient is proportional to the Warburg coefficient (1/A_w_)^2^. The Warburg coefficient (A_w_) of BMO and BMO/C are 260 and 150 Ωs^−1/2^, which confirms that the ionic conductivity of BMO/C is better than that of BMO. 

We also evaluated the electrochemical performance using cyclic voltammograms (CVs). Figure 7c,d show CV curves of the BMO electrode and the BMO/C electrode for the 1st, 2nd and 5th scan, at a scan rate of 0.01 mVs^−1^ in the potential range of 0.01 to 3.0 V vs. Li/Li^+^. It is worth noting that the scan rate is low enough to obtain accurate estimation of the diffusion coefficient. The FWHM of each peak are marked in Figure 7c for BMO and Figure 7d for BMO/C. The FWHM of BMO for the 1st cycle is 0.48 V, while the FWHM of BWO/C is 0.52 V. In addition, after the first oxidation and reduction, the reduction peaks shifted to 0.73 V, indicating the formation of irreversible phase and the reduction of irreversible electrolytes [22]. Normally, the irreversible capacity loss (ICL) is related to the intrinsic properties of metal oxides, the decomposition of electrolyte accompanied by the SEI formation [22]. The FWHM of BMO and BMO/C for the 2nd and 5th are 0.33 V, which corresponded to the stable reversibility of the BMO and BMO/C during the lithium-ion intercalation and deintercalation process. The cathodic peak which appeared at 2.04 V during the first negative scan could be assigned to the formation of Li_x_Bi_2_MoO_6_, as a result of lithium ions inserted into the layered structure of Bi_2_MoO_6_ crystal. The small peak at 1.42 V is related to the reduction of Bi_2_MoO_6_ to Bi and Mo metal [29,31]. During the first cathodic scan, two reduction peaks appeared at 0.52 V and 0.73 V. The obvious cathodic peak at 0.52 V can be associated with the alloying reaction of Bi and Li to form Li_3_Bi [29,31]. Furthermore, the other broad reduction peak is located at 0.73 V, which is related to the two-step formation of LiBi and Li_3_Bi in the electrochemical lithiation reaction process. A strong oxidation peak is observed at 0.98 V, which could be associated with the de-alloying process of Li_3_Bi to Bi [29,31]. Two small anodic peaks appeared at about 1.26 and 2.50 V during charging, which can be attributed to the formation of lithium molybdate and Li_2_O [29,31]. Figure 7d shows the CVs of BMO/C electrode under the same conditions. The results suggest that the anodic peaks of the BMO/C electrode at around 0.52 V in the first sweep move to higher voltages, which could be assigned to the smaller polarization of the BMO/C electrode compared to the BMO electrode. It is worth noting that the peaks of the BMO/C electrode are sharper relative to BMO, and the area between the oxidation peak and the reduction peak is narrower. This indicates that the transfer resistance of Li^+^ in the BMO/C composite is smaller and the rate of redox is higher. Furthermore, the more rounded particle shape of BMO/C is associated with a shorter diffusion length, thus the polarization of BMO/C is weaker than that of BMO. After the first cycle, the reduction and oxidation peaks in the CV curves overlap. This indicates that the electrode shows good reversibility and stability over cycling. The above analysis gives us the basis to postulate the electrochemical reaction mechanism of the BMO electrodes as follows (reactions (1)–(4)).

Bi_2_MoO_6_ + xLi+ +xe-→Li_x_ Bi_2_MoO_6_(1)

Li_x_ Bi_2_MoO_6_ +(12- x)Li+ +(12-x)e-→2Bi+Mo+6Li_2_O(2)

Bi + Li++ e-↔ LiBi(3)

LiBi + 2Li++ 2e-↔Li_3_Bi(4)

### 3.4. Modelling of Li Insertion in the Bi_2_MoO_6_ Lattice

Atomic scale simulations provide a means to evaluate local structure and help to understand the factors influencing the electrochemical behavior of battery materials. Computational techniques have been proven to be very useful for understanding Li incorporation in other oxides previously [70,71,72]. Hence, we have used DFT simulations to model the Li insertion in Bi_2_MoO_6_, to give insights into the local structure properties at the atomic scale. The Bi and Mo layers within the Bi_2_MoO_6_ lattice are strongly bonded, and hence the breaking of this bonding is unlikely to occur [31]. Firstly, substitutional Li defects were investigated by replacing the cations with one Li. The defect formation energy (E_f_) [73] of this substitutional defect (Li_A_) (A = Bi, Mo) was calculated according to
(2)$$E_{f}\left(Li_{A}\right)=E_{Li_{A}}-E_{bulk}+\mu_{A}-\mu_{Li}$$
where $E_{Li_{A}}$ is the total energy of the system with Li on A site, *E_bulk_* is the total energy of the system without the defect, $\mu_{A}$ is the chemical potential of A, and $\mu_{Li}$ the chemical potential of Li. The chemical potentials have been taken as the total energy for a single metal atom from Bi, Mo, and Li metallic bulks, respectively. To evaluate the different inequivalent Bi and Mo lattice sites for Li substitution, the Site-Occupancy Disorder program (SOD) [74] was employed. These inequivalent substitutional lattice sites are shown graphically in Figure 8a–c, respectively. Two inequivalent lattice sites for Bi_Li_ were found, whereas all Mo_Li_ were equivalent. 

Calculating the defect formation energy of Li substitutional defects according to Equation (2), the formation energy for Li substitution on Bi site (Li_Bi_) is 2.31 eV, and 2.35 eV, on site one and two respectively (Figure 8), with the Li on a Mo site (Li_Mo_) giving a much higher defect formation energy of 8.15 eV. It is hence seen that Li substitution of Bi is more energetically favorable than Mo-substitution. Examining the differences in charge density induced by Li substitution (Figure 8), it is seen that both Mo_Li_ and Bi_Li_ cause wide disruption to the charge distribution, with the extra charge density mainly located on the oxygen lattice. Hence, Li substitutional defects could induce the formation of local electric fields, improving the battery performance, but, as these defects have high formation energies, Li substitutional defects are not energetically favorable in the Bi_2_MoO_6_ lattice. Next, Li on interstitial lattice sites is simulated. Interstitial lithium ions in other oxide-based battery materials have previously been postulated by computational and experimental studies [70,72].

For Li interstitial lattice sites, equivalent lithium lattice positions have to be considered. The defect formation energy [73] of Li interstitial in Bi_2_MoO_6_ (Li_int_) was calculated from Equation (3):(3)$$E_{f}\left(Li_{int}\right)=E_{Li_{int}}-E_{bulk}-n\mu_{Li}$$

Here, $E_{Li_{int}}$ is the total energy of the system with the Li interstitial, $E_{bulk}$ is the total energy of Bi_2_MoO_6_ without impurities, *n* is the number of Li atoms in the structure, and $\mu_{Li}$ is the chemical potential of Li as described above. The defect formation energy of a single Li at different interstitial sites in the Bi_2_MoO_6_ lattice is presented in Table 2. 

As opposed to Li substitutional defects, the Li defect formation energy is negative, indicating that Li interstitials are energetically favorable to be present in this system. Li interstitials sites are present in both the MoO_6_-layer and the BiO-layer. Comparing $E_{f}\left(Li_{int}\right),$ the introduction of a single Li interstitial at different lattice sites, it is seen that Li interstitials in the MoO_6_-layer are more energetically favorable than a Li interstitial in the BiO-layer. Examining the difference in lattice response to Li interstitial in terms of charge density (Figure 9), it is clear that Li interstitials in the BiO-layer lead to a larger charge density difference than the equivalent process for a Li interstitial in the MoO_6_-layer. This more concentrated redistribution of charge for the Li interstitial in MoO_6_-layers could be favorable for electrochemical performance, forming a local electric field around the Li interstitial. 

Examining the change in average Bader charges (Table 2) from the pristine Bi_2_MoO_6_ cell to the system with Li interstitials, it is seen that the average Bi, Mo, and O charges remain close to their Bi_2_MoO_6_ values. Examining the change in Bader charge to the species nearest to the Li interstitial site (Table 2), it is observed that upon introducing a Li interstitial in the Bi layer, or in between the layers, the nearest Bi ions to the Li are reduced, as compared to the bulk, by ~0.1 e. Li interstitials in the molybdenum layer do not show any such clear change in the Bi charge state. Finally, in all systems, the standard deviation between the Bi Bader charges is less than 0.03 e, indicating on average no change in the Bi oxidation state. Similar observations are made for the oxygen charges, where a wider charge distribution upon the inclusion of a Li interstitial is seen, centered around the pristine Bi_2_MoO_6_ value. However, it is important to note that slight (<0.1 e) reductions and increases of charge near the lithium interstitial sites occur on the oxygen ions. For Mo, the introduction of Li interstitials to the system does reduce the neighboring Mo ions Bader charge markedly by between 0.05 to 0.2 e. This could indicate a change from Mo^6+^ to Mo^5+^, or a mixed 5+/6+ charge state. Li interstitials in the molybdenum layer also increase the molybdenum Bader charge, by a maximum of 0.06 e, as compared to the pristine bulk. This does not, however, indicate a change to a higher oxidation state [46]. 

Finally, previous computational studies have shown that the DFT-derived defect formation energies can be used to calculate cell voltage trends [75,76,77,78,79,80,81]. The cell voltage vs. Li/Li^+^ (*V*) was calculated using the following formula [76,78,79,82]:(4)$$V=\frac{-E_{f}\left(Li_{int}\right)}{nz}$$
where *n* is the number of lithium interstitial, and *z* is the formal charge of Li. Hence, the cell voltage vs. Li/Li^+^ for the system calculated here for the most favourable Li interstitial site is 2.45 eV. Furthermore, this and all calculated cell voltages sit within the experimental range. 

Following the identification of the MoO_6_-layer as the most favorable layer for Li interstitial sites, the concentration of Li interstitials per molybdenum (x) in the lattice was investigated, up to a total of 1:1 of Li: Mo ratio (x = 1.00). Higher Li: Mo ratios were found to lead to heavy distortions of the Bi_2_MoO_6_ lattice and are hence not presented here. The formation energy $E_{f}\left(Li_{int}\right)\;$ and Li incorporation energy per Li interstitial ($E_{f}\left(Li_{int}\right)/n$) are presented in Table 3. 

From Table 3, it can be seen that the formation of interstitial Li defects is energetically favorable at all x investigated here; the incorporation of Li interstitials in the Bi_2_MoO_6_ lattice is also favourable, in terms of the incorporation energy ($E_{f}\left(Li_{int}\right)/n$) per Li atom. However, it is worth noting that $E_{f}\left(Li_{int}\right)/n$ decreases with increasing x. When examining the electronic structure of Li_x_Bi_2_MoO_6_, it is found that the defect states induced by the Li interstitials are introduced in the band gap. This changes the character of the valence band and conduction band maxima, and thus the Li ion incorporation energy. Nevertheless, the semi-conductor properties of Bi_2_MoO_6_ are maintained. 

## 4. Conclusions

In this work, Bi_2_MoO_6_/carbon (in brief, BMO/C) composite materials have been successfully synthesized by hydrothermal route. XRD and DFT simulations demonstrated that Bi_2_MoO_6_ in the BMO/C composite has the orthorhombic structure. The relative peak intensities of BMO/C are higher than BMO, indicating that the crystallinity of Bi_2_MoO_6_ in the BMO/C composite is increased over that of pure Bi_2_MoO_6_. SEM show Bi_2_MoO rectangle flakes with the size of ~1 μm, whilst the BMO/C composite showed more rounded and smaller (0.2 μm), faceted Bi_2_MoO_6_ flakes, connected with the carbonized palm. The chemical compositions and surface electronic states of the manufactured products were investigated by XPS, confirming the chemical compositions of the materials as expected. The electrochemical performances of Bi_2_MoO_6_ and its composite BMO/C were tested when used as an anode material for Li ion batteries. The initial discharge specific capacity of the BMO/C electrode was 1014 mAh·g^−1^ and remains at 664 mAh·g^−1^ after 200 cycles, much higher than that of Bi_2_MoO_6_. It is worth noting that this good performance of the BMO/C electrode is achieved based on a simple synthesis process and using sustainable low-cost palm carbon, which makes it suitable for large-scale practical applications. 

The Li incorporation mechanism was also estimated by CV for both Bi_2_MoO_6_ and its composite BMO/C. The results indicated that the two materials have the same charge and discharge mechanism. Furthermore, the EIS results show that the BMO/C have lower surface layer resistance, which can improve the electronic conductivity and the electrochemical activity. DFT studies show the fundamental mechanisms of Li ion incorporation into Bi_2_MoO_6_ as Li insertion via the interstitial sites into the MoO_6_-layer. These studies confirm that the improved electrochemical performance of the BMO/C composite is mainly attributed to the enhanced electronic conductivity and more Bi_2_MoO_6_ reaction sites made accessible by palm carbon. All the results show that the BMO/C composite will promote a novel anode material application for Li ion battery.

## Figures and Tables

**Figure 1 materials-13-01132-f001:**
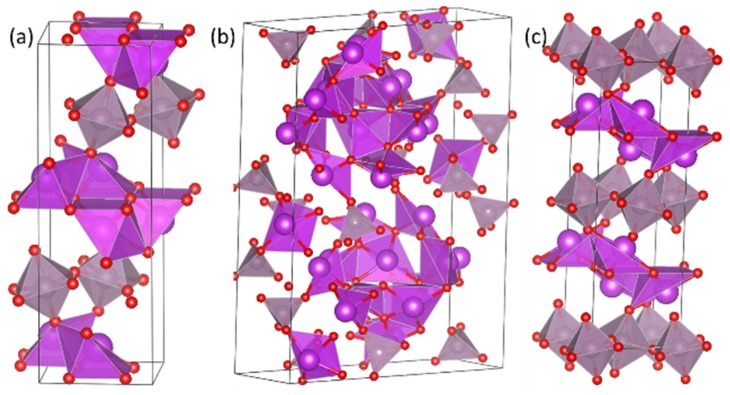
Polyhedral representation of (**a**) orthorhombic Pca2_1_, (**b**) monoclinic p21/c, and (**c**) orthorhombic Pbca structures of Bi_2_MoO_6_. Purple spheres are Bi, grey Mo, and red O.

**Figure 2 materials-13-01132-f002:**
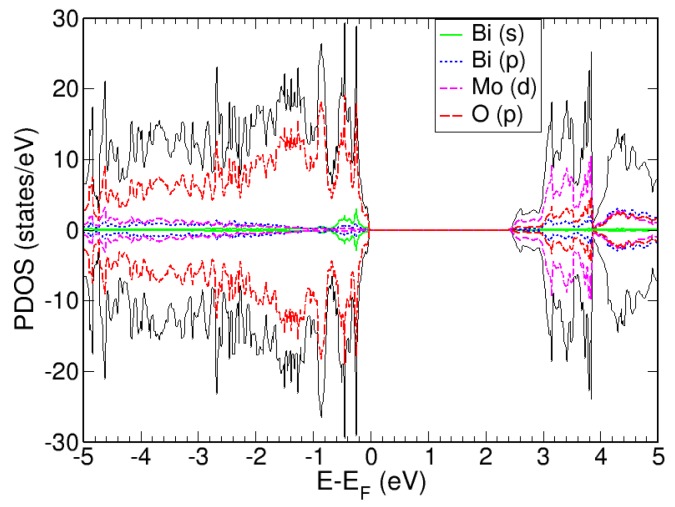
Projected density of states (PDOS) for Bi_2_MoO_6_. Energies (E) on the *x*-axis are referenced to the Fermi level (E_F_). All PDOS below E-E_F_ = 0 eV represent occupied states (valence band), whereas those above E-E_F_ = 0 eV, are unoccupied states (conduction band). In addition, positive PDOS values are the α-spin occupations and negative PDOS values are β-spin occupations.

**Figure 3 materials-13-01132-f003:**
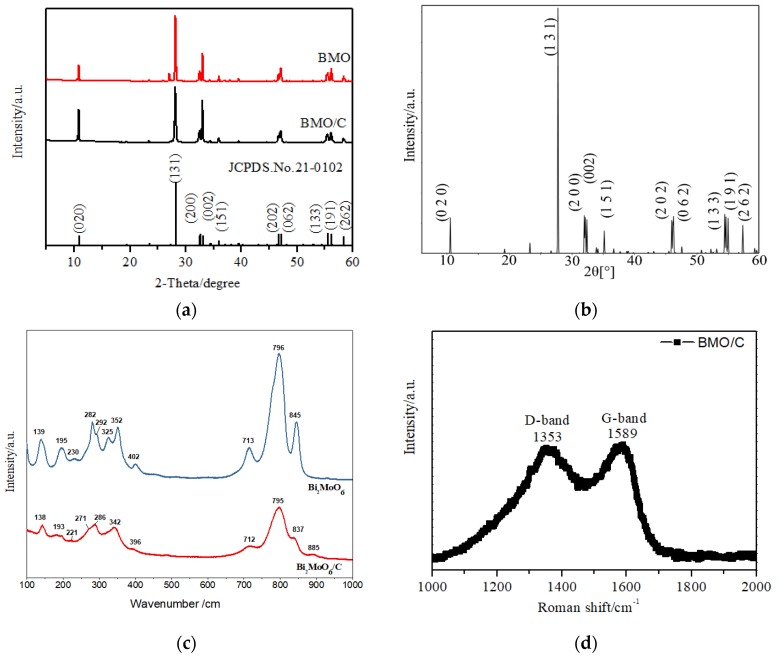
(**a**) XRD patterns of BMO/C, BMO and standard card, and (**b**) DFT simulated XRD pattern for BMO (**c**) Raman spectrum of BMO and BMO/C samples in the wavelength range of 100–1000 cm^−1^; (**d**) Raman spectrum of the as- synthesized BMO/C in the wavelength range of 1000–2000 cm^−1^.

**Figure 4 materials-13-01132-f004:**
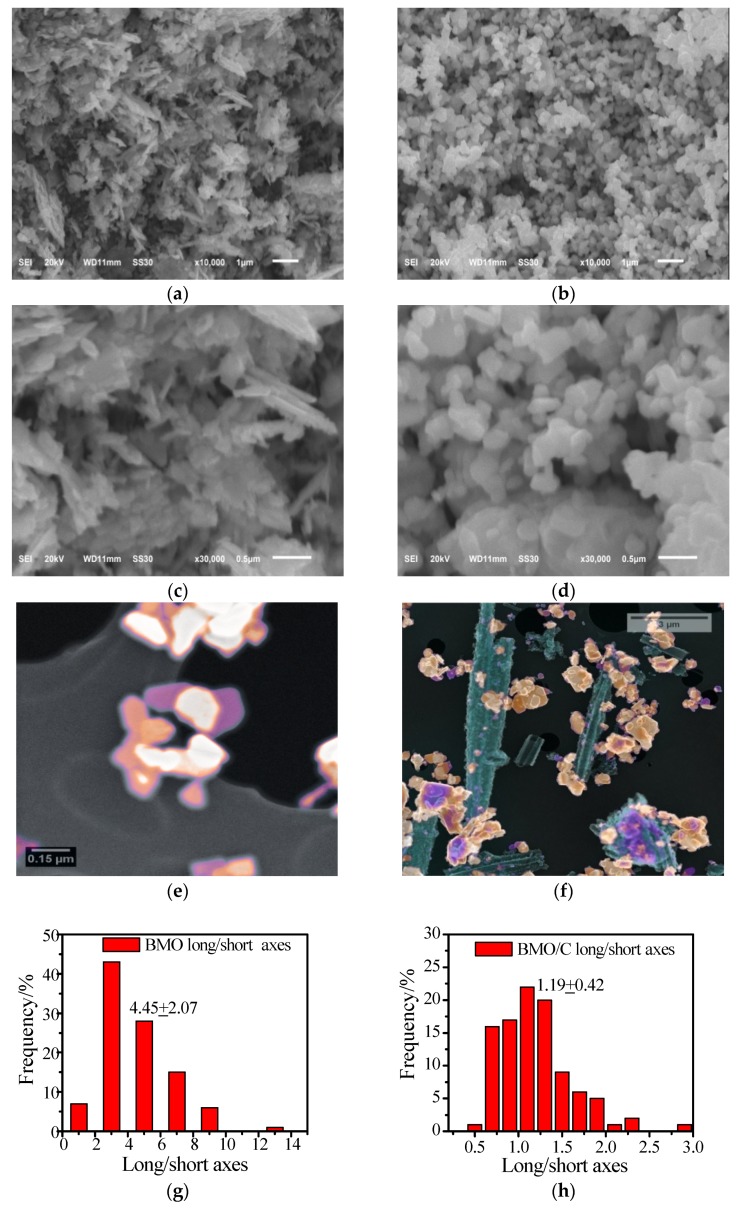
(**a**,**c**) low and high magnification SEM images of BMO; (**b**,**d**) low and high magnification SEM images of BMO/C; (**e**,**f**) composite false-color STEM images of the BMO and BMO/C samples, with the Z-contrast image superimposed on the Secondary electron background image; bright colors signify high Z material (The images have been colorized by assigning the grey scale intensity range to a colour tone range using ImageJ and its look-up-tables Blue-Orange and Thallium for the Z-contrast and the secondary electron image, respectively); (**g**,**h**) statistically calculated long/ short axes ratio of the BMO particles in the BMO and BMO/C samples; (**i**,j) statistically calculated BMO particle size in the BMO and BMO/C samples.

**Figure 5 materials-13-01132-f005:**
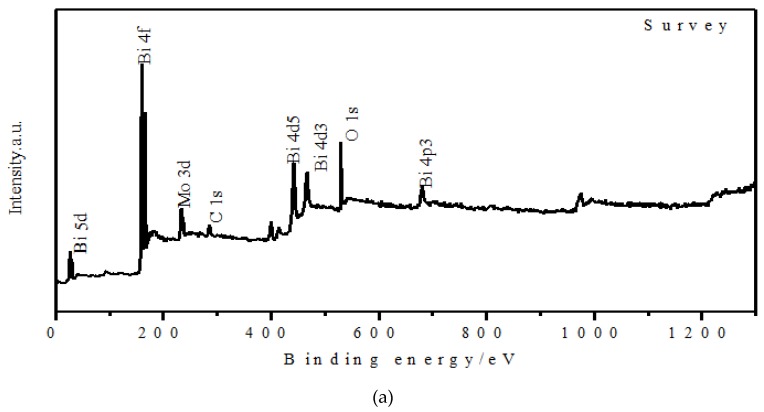
XPS spectra of the BMO/C sample: (**a**) survey, (**b**) Bi 4f, (**c**) Mo 3d, (**d**) C 1s, (**e**) O 1s.

**Figure 6 materials-13-01132-f006:**
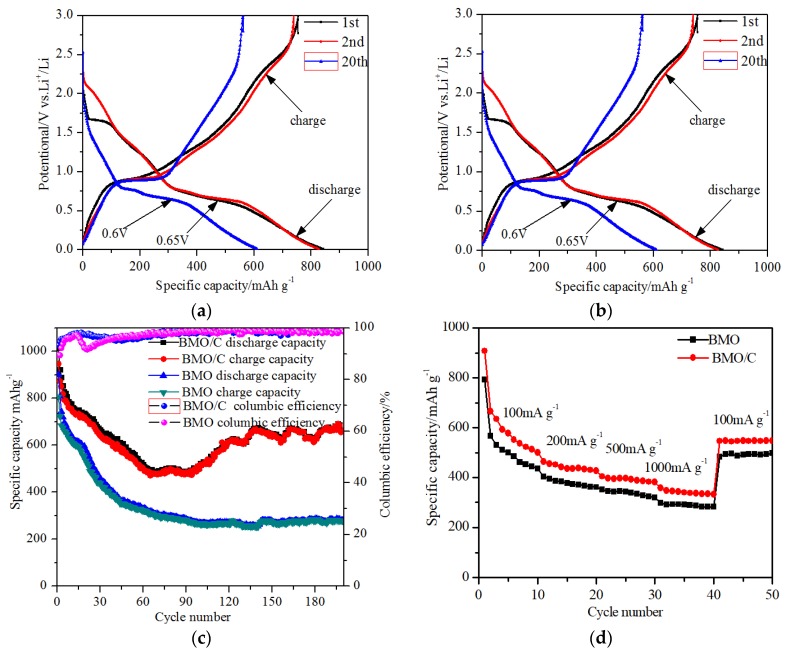
Discharge and charge curves of (**a**) BMO, (**b**) BMO/C electrodes for the 1st, 2nd and 20th cycle at a constant current density 100 mA·g^−1^, and (**c**) cycling performance and the corresponding columbic efficiency of BMO and BMO/C electrodes at current density of 100 mA·g^−1^, (**d**) discharge and charge rate performance of BMO and BMO/C electrodes at current densities of 100, 200, 500, 1000 and 100 mA·g^−1^.

**Figure 7 materials-13-01132-f007:**
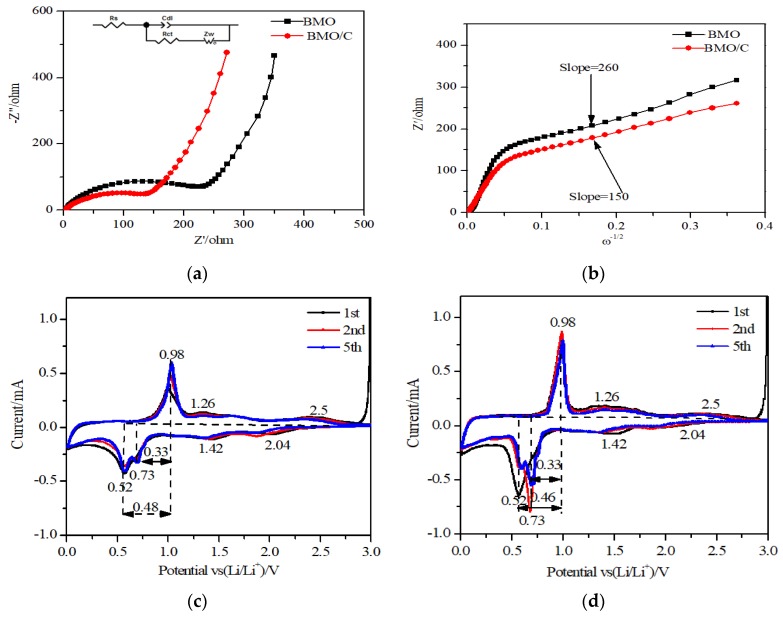
(**a**) electrochemical impedance spectra (EIS) of the BMO and BMO/C; (**b**) linear fitting of Warburg impedance of the BMO and BMO/C electrodes; cyclic voltammograms and FWHM for the 1st, 2nd and 5th cycles of (**c**) BMO and (**d**) BMO/C electrodes at the rate of 0.1 mVs^−1^.

**Figure 8 materials-13-01132-f008:**
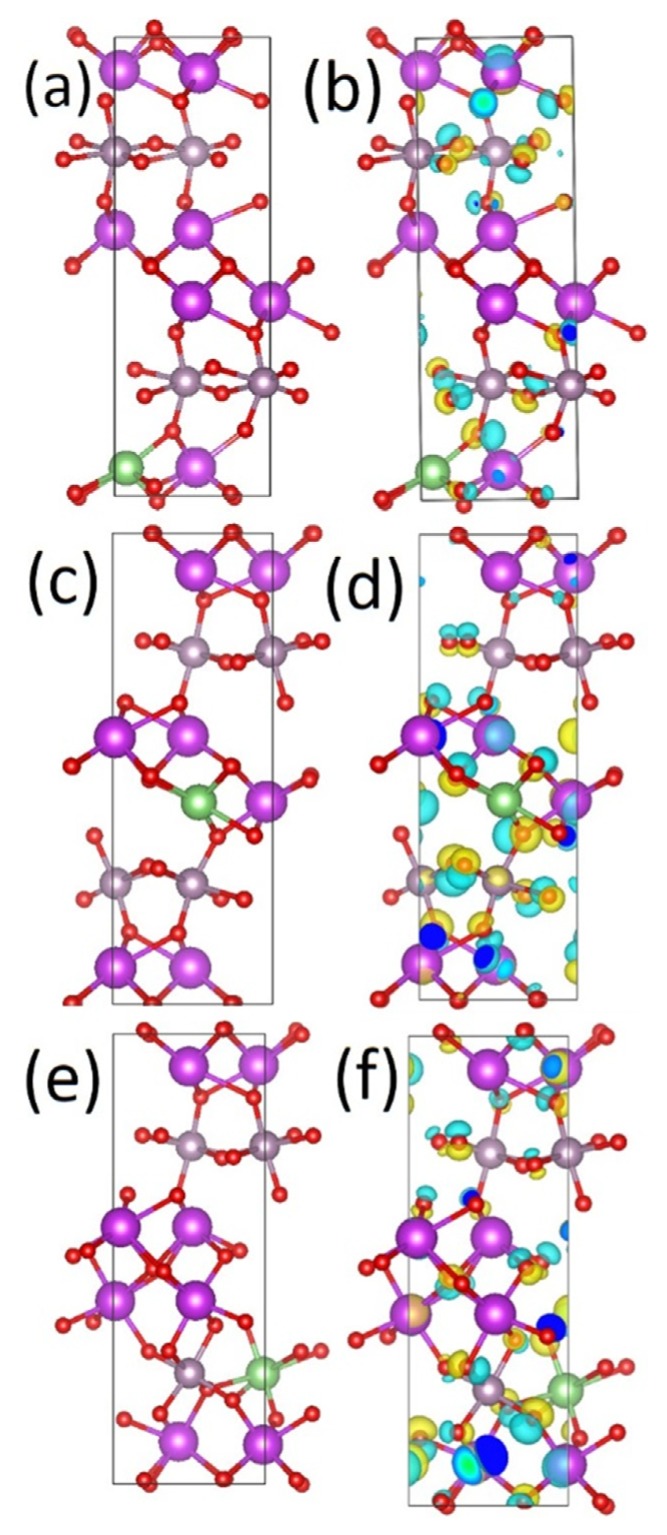
Side view of the optimised Li substitutional defects for (**a**) Li_Bi_ site one, with charge density difference plot in (**b**), (**c**) Li_Bi_ site two, with its corresponding charge density plot in (**d**), and (**e**) Li_Mo_ and its charge density difference plotted in (**f**). Purple spheres are Bi, grey Mo, green Li, and red O. Yellow iso-surface represents an increase in charge density upon lithium substitution, whereas blue represents depletion of charge density.

**Figure 9 materials-13-01132-f009:**
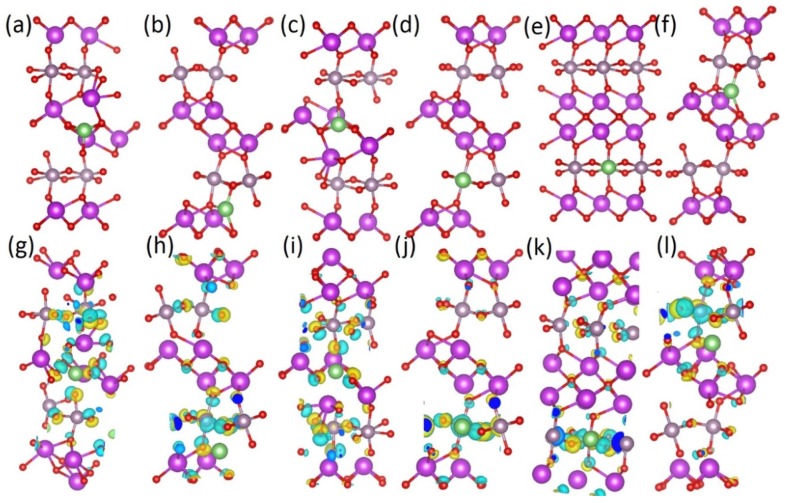
Calculated atomic scale models of Li interstitial in Bi_2_MoO_6_ at (**a**) site 1, (**b**) site 2, (**c**) site 3, (**d**) site 4, (**e**) site 5, and (**f**) site 6, and charge density difference as a result of Li interstitial at (**g**) site 1, (**h**) site 2, (**i**) site 3, (**j**) site 4, (**k**) site 5, and (**l**) site 6 in Bi_2_MoO_6_. The site numbering refers to Table 2. Purple spheres are Bi, grey Mo, green Li, and red O. Yellow iso-surface represents an increase in charge density upon lithium substitution, whereas blue iso-surface represents depletion of charge density.

**Table 1 materials-13-01132-t001:** Comparison of calculated lattice vectors (*a*, *b*, and *c*), angles (α, β, and γ), and band gap.

	*a* (Å)	*b* (Å)	*c* (Å)	α = β=γ (°)	E_g_ (eV)
Experimental	5.45	16.47	5.47	90	2.53 [53], 2.56 [54]
GGA	5.658	16.491	5.676	90	2.23
GGA + U	5.448	16.506	5.467	90	2.45

**Table 2 materials-13-01132-t002:** Defect formation energy (E_f_) in eV, average Bader charges (q) with standard deviation, and unique Bader charge for nearest neighbor (NN) atoms to Li site (in e) for insertion of a single Li at different lattice sites. Graphical representation of interstitial site numbering is presented in Figure 9. For comparison, q_Bi_ = 1.91 e, q_Mo_ = 2.76 e, and q_O_= −1.10 e for pristine Bi_2_MoO_6_.

Site	*E_f_*	q_Bi_	q_Bi,NN_	q_Mo_	q_Mo,NN_	q_O_	q_O,NN_	q_Li_
**BiO-layer**								
**1**	−1.79	1.85 ± 0.03	1.83, 1.85	2.68 ± 0.08	2.55, 2.67	−1.08 ± 0.05	−1.05, −1.11, −1.10	0.51
**2**	−1.44	1.83 ± 0.02	1.79	2.64 ± 0.03	2.60, 2.67	−1.07 ± 0.05	−1.03, −1.08, −1.15	0.52
**3**	−1.79	1.85±0.03	1.80, 1.82	2.70±0.08	2.57, 2.73	−1.09 ± 0.04	−1.12, −1.07, −1.11	0.62
**MoO_6_-layer**								
**4**	−2.34	1.87 ± 0.02	1.89, 1.85	2.75 ± 0.06	2.66, 2.82	−1.11 ± 0.04	−1.08, −1.15	0.63
**5**	−2.45	1.88 ± 0.02	1.87	2.72 ± 0.07	2.62, 2.80	−1.10 ± 0.04	−1.04, −1.09	0.61
In-between layers								
**6**	−1.45	1.84 ± 0.02	1.81, 1.82	2.61 ± 0.09	2.70, 2.46	−1.07 ± 0.05	−1.09, −1.10	0.54

**Table 3 materials-13-01132-t003:** Formation energy of a Li interstitial defect, and Li incorporation energy for different numbers of Li (n_Li_) in lattice.

n_Li_	$E_{f}\left(Li_{int}\right)\;(\mathrm{eV})$	*E_incorp_* (eV/Li)
**1**	−2.45	−2.45
**2**	−4.38	−2.19
**3**	−6.58	−2.19
**4**	−7.79	−1.95

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
