# Peer review of "Synthesis and Electrochemical Properties of Bi2MoO6/Carbon Anode for Lithium-Ion Battery Application"

_materials, 2020, doi:10.3390/ma13051132_

Round 1

Reviewer 1 Report

Synthesis and electrochemical properties of 1 Bi2MoO6/Carbon anode for lithium-ion battery 2 application

Zhang etal.

Authors prepared Bi2MoO6/carbon anode materials and studied its electrochemical properties and validated with computation method . The following minor revisions and additional studies are needed to reconsider above paper in “Materials ‘

Introduction it will be nice to discuss few anodes martials recent review papers ex ; Ref. Chemical Reviews 113(2013)5364

Please mention annealing carriedout in air or Argon, authors noted any evaporation of Bi Please include the lattice fitted lattice parameter values in 3rd decimal values with errors, a, b, c are in italics

Please mention possible difference in the CV and GC profiles shapes by considering ref.40, expand possible reasons like reaction conditions and preparation temp etc .

 7 : effect of scan rate nice to expand more with scan rate, above 5 mV/sec scan rate , just to see the trend and also to discuss with literature observation

This paper is mainly on Mo-based anodes and MO6 metal cluster compounds, it will be nice to discuss previous studies on Mo-based oxides ex : RSC Advances, 6 (2016) 55167, Electrochimica Acta 178(2015)699, RSC Advances 4(2014)33883, Electrochimica Acta 54(2009)3360.

Section 3.2: Since this electroanalytical chemistry journal it will be nice to discuss the possible reasons for Irreversible capacity loss (ICL) occurs in many reasons: like nature of crystal structure, particle size

Also Impedance studies it not clear at what voltage EIS studies are taken, please include fresh or what voltage and discharge or charge cycle. it will be nice include the voltage are recorded and EIS values are sensitive voltage. Please read important references and give the fitted impedance values, give circuit and compare present values with literature:

It will be nice if any additional data on impedance

Reviewer 2 Report

Listing the initial discharge capacity is misleading. It is always larger than any steady state capacity because it includes irreversible side reactions. This value should be removed from the abstract and not used as the topline number in discussion. Several references did not properly show up in the text in section 3.1.1 The authors’ split of Raman data into two separate graphs in Figure 3c-d is confusing. They start and end at the same spot (1000 cm-1), so they should show the continuous spectrum for both plots. Also, for analysis of carbon material, inclusion of the resonant region (2D, ~2700 cm-1) is important. The authors’ claim that the material is highly graphitic is not supported by the data. The D and G bands are broad, and there is no information from the resonant modes. What is the weight ratio of metal oxide to carbon in the synthesized structures? Data in Figure 6c is clearly flawed. Capacity should not rise on its own with high cycle numbers. This shows that the device is unstable or was not properly pre-cycled. The high charge transfer resistance (size of semicircle) in Figure 7a shows poor electrode design. Neither 138 nor 246 ohms are good numbers.

Reviewer 3 Report

This manuscript reports on the synthesis and electrochemical properties of Bi2MoO6/Carbon anode for lithium-ion batteries. By incorporating the palm carbon, the BMO/C electrode showed better performance. To confirm the experimental data, various simulations were done. It is interesting. However, there are many issues to be addressed.

Please explain why the carbon source was oxidized and then carbonized. Is the prepared carbon source well dispersed in DI water? The authors showed the Raman data separately (Fig. 3c and 3d). It is much better to show the Raman data in one graph to compare between BMO and BMO/C samples. Furthermore, when looking into the D and G values, the ID/IG is approximately 0.85, not 0.97. It should be rechecked. Line 275~276: the explanation on the particle size should be rechecked. There are several reported BMO/carbon electrodes for LIBs. It is necessary to compare in order to exhibit an advance of the developed material. There are many redundant sentences, phrases, and typos. Line 75: the phrase “high specific surface area” is written twice. Line 100-102 on page 3: section 2.1: following sentence was repeatedly written: “The Bi2MoO6 material was synthesized by hydrothermal methods. 2 mmol Bi(NO3)3 and 1 mmol Na2MoO4 were dissolved in 20 ml de-ionized water under magnetically stirring for 1 hour” and “The Bi2MoO6/carbon composite was synthesized by hydrothermal methods. In a typical procedure, bismuth nitrate and sodium molybdate, with a Bi/Mo molar ratio of 2:1, were dissolved in 20 ml de-ionized water, under magnetic stirring” (Line 113-115). In section 3.1.1: the characteristics of the DFT was redundantly mentioned: “(DFT) is extremely powerful tool”….(Line 171) and “DFT simulations have proved to be a powerful tool in the battery materials community”…in same paragraph (line 176). In line 177, “Here, firstly the pristine Bi2MoO6 structure before Li addition is studied to understand the atomic scale structure of our anode material…” and In line 179 “As a first step, we have investigated the crystal structure of Bi2MoO6” These are redundant. In line 285: ”of the” is repeated In line 291: ”and BMO” should be deleted. In line 197, mentioned references [49] and [50] are not matched those in Table 1. In line 227, the authors should clearly indicate which SEM and TEM images are. In line 229, 245: the mentioned figure numbers are wrong. In Fig.(4): the figure for STEM of BMO/C is not labeled.

Round 2

Reviewer 2 Report

The authors did not address the concerns in the original review.

The first comment stated that the 664 mAh/g value was misleading. The authors ignored this comment and failed to address it or correct the manuscript in any way.

The reference errors on pages 4-5 are still there.

The cycling data is still flawed. Any data that shows capacitance improving with cycling (and thus activation) is flawed data, and the authors cannot fall on that as justification for their work. If precycling is activating the material, the data collected from it cannot be considered as legitimate electrochemical data.

Author Response

We thank the Reviewer for the careful reading of our paper and for the helpful comments. We have addressed these comments, as in the attached response file.

Reviewer 3 Report

The authors revised the manuscript based on the reviewer's comments. It can be published to Materials.

Author Response

We thank the Reviewer for the support for publishing our paper.